# Covariance-Controlled Adaptive Langevin Thermostat for Large-Scale Bayesian Sampling

**Xiaocheng Shang**[*]
University of Edinburgh
x.shang@ed.ac.uk

**Zhanxing Zhu**[*]
University of Edinburgh
zhanxing.zhu@ed.ac.uk

**Benedict Leimkuhler**
University of Edinburgh
b.leimkuhler@ed.ac.uk

**Amos J. Storkey**
University of Edinburgh
a.storkey@ed.ac.uk

## Abstract

Monte Carlo sampling for Bayesian posterior inference is a common approach used in machine learning. The Markov Chain Monte Carlo procedures that are used are often discrete-time analogues of associated stochastic differential equations (SDEs). These SDEs are guaranteed to leave invariant the required posterior distribution. An area of current research addresses the computational benefits of stochastic gradient methods in this setting. Existing techniques rely on estimating the variance or covariance of the subsampling error, and typically assume constant variance. In this article, we propose a covariance-controlled adaptive Langevin thermostat that can effectively dissipate parameter-dependent noise while maintaining a desired target distribution. The proposed method achieves a substantial speedup over popular alternative schemes for large-scale machine learning applications.

## 1 Introduction

In machine learning applications, direct sampling with the entire large-scale dataset is computationally infeasible. For instance, standard Markov Chain Monte Carlo (MCMC) methods [16], as well as typical Hybrid Monte Carlo (HMC) methods [3, 6, 9], require the calculation of the acceptance probability and the creation of informed proposals based on the whole dataset.

In order to improve computational efficiency, a number of stochastic gradient methods [4, 5, 20, 21] have been proposed in the setting of Bayesian sampling based on random (and much smaller) subsets to approximate the likelihood of the whole dataset, thus substantially reducing the computational cost in practice. Welling and Teh proposed the so-called Stochastic Gradient Langevin Dynamics (SGLD) [21], combining the ideas of stochastic optimization [18] and traditional Brownian dynamics, with a sequence of stepsizes decreasing to zero. A fixed stepsize is often adopted in practice which is the choice in this article as in Vollmer et al. [20], where a modified SGLD (mSGLD) was also introduced that was designed to reduce sampling bias.

SGLD generates samples from first order Brownian dynamics, and thus, with a fixed timestep, one can show that it is unable to dissipate excess noise in gradient approximations while maintaining the desired invariant distribution [4]. A Stochastic Gradient Hamiltonian Monte Carlo (SGHMC) method was proposed by Chen et al. [4], which relies on second order Langevin dynamics and incorporates a parameter-dependent diffusion matrix that is intended to effectively offset the stochastic perturbation of the gradient. However, it is difficult to accommodate the additional diffusion term

---

[*]The first and second authors contributed equally, and the listed author order was decided by lot.

in practice. Moreover, as pointed out in [5] poor estimation of it may have a significant adverse influence on the sampling of the target distribution; for example the effective system temperature may be altered.

The "thermostat" idea, which is widely used in molecular dynamics [7, 13], was recently adopted in the Stochastic Gradient Nosé-Hoover Thermostat (SGNHT) by Ding et al. [5] in order to adjust the kinetic energy during simulation in such a way that the canonical ensemble is preserved (i.e. so that a prescribed constant temperature distribution is maintained). In fact, the SGNHT method is essentially equivalent to the Adaptive Langevin (Ad-Langevin) thermostat proposed earlier by Jones and Leimkuhler [10] in the molecular dynamics setting (see [15] for discussion).

Despite the substantial interest generated by these methods, the mathematical foundation for stochastic gradient methods has been incomplete. The underlying dynamics of the SGNHT [5] was taken up by Leimkuhler and Shang [15], together with the design of discretization schemes with high effective order of accuracy. SGNHT methods are designed based on the assumption of constant noise variance. In this article, we propose a Covariance-Controlled Adaptive Langevin (CCAdL) thermostat, that can handle parameter-dependent noise, improving both robustness and reliability in practice, and which can effectively speed up the convergence to the desired invariant distribution in large-scale machine learning applications.

The rest of the article is organized as follows. In Section 2, we describe the setting of Bayesian sampling with noisy gradients and briefly review existing techniques. In Section 3, we construct the novel Covariance-Controlled Adaptive Langevin (CCAdL) method that can effectively dissipate parameter-dependent noise while maintaining the correct distribution. Various numerical experiments are performed in Section 4 to verify the usefulness of CCAdL in a wide range of large-scale machine learning applications. Finally, we summarize our findings in Section 5.

## 2 Bayesian Sampling with Noisy Gradients

In the typical setting of Bayesian sampling [3, 19], one is interested in drawing states from a posterior distribution defined as

$$\pi(\boldsymbol{\theta}|\mathbf{X}) \propto \pi(\mathbf{X}|\boldsymbol{\theta})\pi(\boldsymbol{\theta}), \tag{1}$$

where $\boldsymbol{\theta} \in \mathbb{R}^{N_\mathrm{d}}$ is the parameter vector of interest, $\mathbf{X}$ denotes the entire dataset, and, $\pi(\mathbf{X}|\boldsymbol{\theta})$ and $\pi(\boldsymbol{\theta})$ are the likelihood and prior distributions, respectively. We introduce a potential energy function $U(\boldsymbol{\theta})$ by defining $\pi(\boldsymbol{\theta}|\mathbf{X}) \propto \exp(-\beta U(\boldsymbol{\theta}))$, where $\beta$ is a positive parameter and can be interpreted as being proportional to the reciprocal temperature in an associated physical system, i.e. $\beta^{-1} = k_\mathrm{B}T$ ($k_\mathrm{B}$ is the Boltzmann constant and $T$ is temperature). In practice, $\beta$ is often set to be unity for notational simplicity. Taking the logarithm of (1) yields

$$U(\boldsymbol{\theta}) = -\log \pi(\mathbf{X}|\boldsymbol{\theta}) - \log \pi(\boldsymbol{\theta}). \tag{2}$$

Assuming the data are independent and identically distributed (i.i.d.), the logarithm of the likelihood can be calculated as

$$\log \pi(\mathbf{X}|\boldsymbol{\theta}) = \sum_{i=1}^{N} \log \pi(\mathbf{x}_i|\boldsymbol{\theta}), \tag{3}$$

where $N$ is the size of the entire dataset.

However, as already mentioned, it is computationally infeasible to deal with the entire large-scale dataset at each timestep as would typically be required in MCMC and HMC methods. Instead, in order to improve the efficiency, a random (and much smaller, $n \ll N$) subset is preferred in stochastic gradient methods, in which the likelihood of the dataset for given parameters is approximated as

$$\log \pi(\mathbf{X}|\boldsymbol{\theta}) \approx \frac{N}{n} \sum_{i=1}^{n} \log \pi(\mathbf{x}_{r_i}|\boldsymbol{\theta}), \tag{4}$$

where $\{\mathbf{x}_{r_i}\}_{i=1}^{n}$ represents a random subset of $\mathbf{X}$. Thus, the "noisy" potential energy can be written as

$$\tilde{U}(\boldsymbol{\theta}) = -\frac{N}{n} \sum_{i=1}^{n} \log \pi(\mathbf{x}_{r_i}|\boldsymbol{\theta}) - \log \pi(\boldsymbol{\theta}), \tag{5}$$

where the negative gradient of the potential is referred to as the "noisy" force, i.e. $\tilde{\mathbf{F}}(\boldsymbol{\theta}) = -\nabla \tilde{U}(\boldsymbol{\theta})$.

Our goal is to correctly sample the Gibbs distribution $\rho(\boldsymbol{\theta}) \propto \exp(-\beta U(\boldsymbol{\theta}))$ (1). As in [4, 5], the gradient noise is assumed to be Gaussian with mean zero and unknown variance, in which case one may rewrite the noisy force as

$$\tilde{\mathbf{F}}(\boldsymbol{\theta}) = -\nabla U(\boldsymbol{\theta}) + \sqrt{\boldsymbol{\Sigma}(\boldsymbol{\theta})}\mathbf{M}^{1/2}\mathbf{R}\,, \qquad (6)$$

where $\mathbf{M}$ typically is a diagonal matrix, $\boldsymbol{\Sigma}(\boldsymbol{\theta})$ represents the covariance matrix of the noise and $\mathbf{R}$ is a vector of i.i.d. standard normal random variables. Note that $\sqrt{\boldsymbol{\Sigma}(\boldsymbol{\theta})}\mathbf{M}^{1/2}\mathbf{R}$ here is actually equivalent to $\mathcal{N}\left(\mathbf{0}, \boldsymbol{\Sigma}(\boldsymbol{\theta})\mathbf{M}\right)$.

In a typical setting of numerical integration with associated stepsize $h$, one has

$$h\tilde{\mathbf{F}}(\boldsymbol{\theta}) = h\left(-\nabla U(\boldsymbol{\theta}) + \sqrt{\boldsymbol{\Sigma}(\boldsymbol{\theta})}\mathbf{M}^{1/2}\mathbf{R}\right) = -h\nabla U(\boldsymbol{\theta}) + \sqrt{h}\left(\sqrt{h\boldsymbol{\Sigma}(\boldsymbol{\theta})}\right)\mathbf{M}^{1/2}\mathbf{R}\,, \qquad (7)$$

and therefore, assuming a constant covariance matrix (i.e. $\boldsymbol{\Sigma} = \sigma^2\mathbf{I}$, where $\mathbf{I}$ is the identity matrix), the SGNHT method by Ding et al. [5], has the following underlying dynamics, written as a standard Itō stochastic differential equation (SDE) system [15]:

$$\mathrm{d}\boldsymbol{\theta} = \mathbf{M}^{-1}\mathbf{p}\,\mathrm{d}t\,,$$
$$\mathrm{d}\mathbf{p} = -\nabla U(\boldsymbol{\theta})\mathrm{d}t + \sigma\sqrt{h}\mathbf{M}^{1/2}\mathrm{d}\mathbf{W} - \xi\mathbf{p}\,\mathrm{d}t + \sqrt{2A\beta^{-1}}\mathbf{M}^{1/2}\mathrm{d}\mathbf{W}_{\mathrm{A}}\,, \qquad (8)$$
$$\mathrm{d}\xi = \mu^{-1}\left[\mathbf{p}^T\mathbf{M}^{-1}\mathbf{p} - N_{\mathrm{d}}k_{\mathrm{B}}T\right]\mathrm{d}t\,,$$

where, colloquially, $\mathrm{d}\mathbf{W}$ and $\mathrm{d}\mathbf{W}_{\mathrm{A}}$, respectively, represent vectors of independent Wiener increments; and are often informally denoted by $\mathcal{N}(\mathbf{0}, \mathrm{d}t\mathbf{I})$ [4]. The coefficient $\sqrt{2A\beta^{-1}}\mathbf{M}^{1/2}$, represents the strength of artificial noise added into the system to improve ergodicity, and $A$, which can be termed as "effective friction", is a positive parameter and proportional to the variance of the noise. The auxiliary variable $\xi \in \mathbb{R}$ is governed by a Nosé-Hoover device [8, 17] via a negative feedback mechanism, i.e. when the instantaneous temperature (average kinetic energy per degree of freedom) calculated as

$$k_{\mathrm{B}}T = \mathbf{p}^T\mathbf{M}^{-1}\mathbf{p}/N_{\mathrm{d}} \qquad (9)$$

is below the target temperature, the "dynamical friction" $\xi$ would decrease allowing an increase of temperature, while $\xi$ would increase when the temperature is above the target. $\mu$ is a coupling parameter which is referred to as the "thermal mass" in the molecular dynamics setting.

**Proposition 1:** (See Jones and Leimkuhler [10]) The SGNHT method (8) preserves the modified Gibbs (stationary) distribution

$$\tilde{\rho}_\beta(\boldsymbol{\theta}, \mathbf{p}, \xi) = Z^{-1}\exp\left(-\beta H(\boldsymbol{\theta}, \mathbf{p})\right)\exp\left(-\beta\mu(\xi - \bar{\xi})^2/2\right)\,, \qquad (10)$$

where $Z$ is the normalizing constant, $H(\boldsymbol{\theta}, \mathbf{p}) = \mathbf{p}^T\mathbf{M}^{-1}\mathbf{p}/2 + U(\boldsymbol{\theta})$ is the Hamiltonian, and

$$\bar{\xi} = A + \beta h\sigma^2/2\,. \qquad (11)$$

Proposition 1 tells us that the SGNHT method can adaptively dissipate excess noise pumped into the system while maintaining the correct distribution. The variance of the gradient noise, $\sigma^2$, does not need to be known a priori. As long as $\sigma^2$ is constant, the auxiliary variable $\xi$ will be able to automatically find its mean value $\bar{\xi}$ on the fly. However, with a parameter-dependent covariance matrix $\boldsymbol{\Sigma}(\boldsymbol{\theta})$, the SGNHT method (8) would not produce the required target distribution (10).

Ding et al. [5] claimed that it is reasonable to assume the covariance matrix $\boldsymbol{\Sigma}(\boldsymbol{\theta})$ is constant when the size of the dataset, $N$, is large, in which case the variance of the posterior of $\boldsymbol{\theta}$ is small. The magnitude of the posterior variance does not actually relate to the constancy of the $\boldsymbol{\Sigma}$, however, in general $\boldsymbol{\Sigma}$ is not constant. Simply assuming the non-constancy of the $\boldsymbol{\Sigma}$ can have a significant impact on the performance of the method (most notably the stability measured by the largest usable stepsize). Therefore, it is essential to have an approach that can handle parameter-dependent noise. In the following section we propose a covariance-controlled thermostat that can effectively dissipate parameter-dependent noise while maintaining the target stationary distribution.

## 3 Covariance-Controlled Adaptive Langevin Thermostat

As mentioned in the previous section, the SGNHT method (8) can only dissipate noise with a constant covariance matrix. When the covariance matrix becomes parameter-dependent, in general a parameter-dependent covariance matrix does not imply the required "thermal equilibrium", i.e. the system cannot be expected to converge to the desired invariant distribution (10), typically resulting in poor estimation of functions of parameters of interest. In fact, in that case it is not clear whether or not there exists an invariant distribution at all.

In order to construct a stochastic-dynamical system that preserves the canonical distribution, we suggest adding a suitable damping (viscous) term to effectively dissipate the parameter-dependent gradient noise. To this end, we propose the following Covariance-Controlled Adaptive Langevin (CCAdL) thermostat

$$\mathrm{d}\boldsymbol{\theta} = \mathbf{M}^{-1}\mathbf{p}\mathrm{d}t\,,$$

$$\mathrm{d}\mathbf{p} = -\nabla U(\boldsymbol{\theta})\mathrm{d}t + \sqrt{h\boldsymbol{\Sigma}(\boldsymbol{\theta})}\mathbf{M}^{1/2}\mathrm{d}\mathbf{W} - (h/2)\beta\boldsymbol{\Sigma}(\boldsymbol{\theta})\mathbf{p}\mathrm{d}t - \xi\mathbf{p}\mathrm{d}t + \sqrt{2A\beta^{-1}}\mathbf{M}^{1/2}\mathrm{d}\mathbf{W}_{\mathrm{A}}\,, \quad (12)$$

$$\mathrm{d}\xi = \mu^{-1}\left[\mathbf{p}^T\mathbf{M}^{-1}\mathbf{p} - N_{\mathrm{d}}k_{\mathrm{B}}T\right]\mathrm{d}t\,.$$

**Proposition 2:** The CCAdL thermostat (12) preserves the modified Gibbs (stationary) distribution

$$\hat{\rho}_\beta(\boldsymbol{\theta},\mathbf{p},\xi) = Z^{-1}\exp\left(-\beta H(\boldsymbol{\theta},\mathbf{p})\right)\exp\left(-\beta\mu(\xi-A)^2/2\right)\,. \quad (13)$$

*Proof:* The Fokker-Planck equation corresponding to (12) is

$$\rho_t = \mathcal{L}^\dagger\rho := -\mathbf{M}^{-1}\mathbf{p}\cdot\nabla_{\boldsymbol{\theta}}\rho + \nabla U(\boldsymbol{\theta})\cdot\nabla_{\mathbf{p}}\rho + (h/2)\nabla_{\mathbf{p}}\cdot(\boldsymbol{\Sigma}(\boldsymbol{\theta})\mathbf{M}\nabla_{\mathbf{p}}\rho) + (h/2)\beta\nabla_{\mathbf{p}}\cdot(\boldsymbol{\Sigma}(\boldsymbol{\theta})\mathbf{p}\rho)$$

$$+ \xi\nabla_{\mathbf{p}}\cdot(\mathbf{p}\rho) + A\beta^{-1}\nabla_{\mathbf{p}}\cdot(\mathbf{M}\nabla_{\mathbf{p}}\rho) - \mu^{-1}\left[\mathbf{p}^T\mathbf{M}^{-1}\mathbf{p} - N_{\mathrm{d}}k_{\mathrm{B}}T\right]\nabla_\xi\rho\,.$$

Just insert $\hat{\rho}_\beta$ (13) into the Fokker-Planck operator $\mathcal{L}^\dagger$ to see that it vanishes. $\qquad\square$.

The incorporation of the parameter-dependent covariance matrix $\boldsymbol{\Sigma}(\boldsymbol{\theta})$ in (12) is intended to offset the covariance matrix coming from the gradient approximation. However, in practice, one does not know $\boldsymbol{\Sigma}(\boldsymbol{\theta})$ a priori. Thus instead one must estimate $\boldsymbol{\Sigma}(\boldsymbol{\theta})$ during the simulation, a task which will be addressed in Section 3.1. This procedure is related to the method used in the SGHMC method proposed by Chen et al. [4], which uses dynamics of the following form:

$$\mathrm{d}\boldsymbol{\theta} = \mathbf{M}^{-1}\mathbf{p}\mathrm{d}t\,,$$

$$\mathrm{d}\mathbf{p} = -\nabla U(\boldsymbol{\theta})\mathrm{d}t + \sqrt{h\boldsymbol{\Sigma}(\boldsymbol{\theta})}\mathbf{M}^{1/2}\mathrm{d}\mathbf{W} - A\mathbf{p}\mathrm{d}t + \sqrt{2\beta^{-1}\left(A\mathbf{I} - h\boldsymbol{\Sigma}(\boldsymbol{\theta})/2\right)}\mathbf{M}^{1/2}\mathrm{d}\mathbf{W}_{\mathrm{A}}\,. \quad (14)$$

It can be shown that the SGHMC method preserves the Gibbs canonical distribution

$$\rho_\beta(\boldsymbol{\theta},\mathbf{p}) = Z^{-1}\exp\left(-\beta H(\boldsymbol{\theta},\mathbf{p})\right)\,. \quad (15)$$

Although both CCAdL (12) and SGHMC (14) preserve their respective invariant distributions, let us note several advantages of the former over the latter in practice:

(i) CCAdL and SGHMC both require estimation of the covariance matrix $\boldsymbol{\Sigma}(\boldsymbol{\theta})$ during simulation, which can be costly in high dimension. In numerical experiments, we have found that simply using the diagonal of the covariance matrix, at significantly reduced computational cost, works quite well in CCAdL. By contrast, it is difficult to find a suitable value of the parameter $A$ in SGHMC since one has to make sure the matrix $A\mathbf{I} - h\boldsymbol{\Sigma}(\boldsymbol{\theta})/2$ is positive semi-definite. One may attempt to use a large value of the "effective friction" $A$ and/or a small stepsize $h$. However, too-large a friction would essentially reduce SGHMC to SGLD, which is not desirable, as pointed out in [4], while extremely small stepsize would significantly impact the computational efficiency.

(ii) Estimation of the covariance matrix $\boldsymbol{\Sigma}(\boldsymbol{\theta})$ unavoidably introduces additional noise in both CCAdL and SGHMC. Nonetheless, CCAdL can still effectively control the system temperature (i.e. maintaining the correct distribution of the momenta) due to the use of the stabilizing Nosé-Hoover control, while in SGHMC poor estimation of the covariance matrix may lead to significant deviations of the system temperature (as well as the distribution of the momenta), resulting in poor sampling of the parameters of interest.

### 3.1 Covariance Estimation of Noisy Gradients

Under the assumption that the noise of the stochastic gradient follows a normal distribution, we apply a similar method to that of [2] to estimate the covariance matrix associated with the noisy gradient. If we let $g(\boldsymbol{\theta};\mathbf{x}) = \nabla_{\boldsymbol{\theta}}\log\pi(\mathbf{x}|\boldsymbol{\theta})$ and assume that the size of subset $n$ is large enough for the central limit theorem to hold, we have

$$\frac{1}{n}\sum_{i=1}^n g(\boldsymbol{\theta}_t;\mathbf{x}_{r_i}) \sim \mathcal{N}\left(\mathbb{E}_{\mathbf{x}}[g(\boldsymbol{\theta}_t;\mathbf{x})], \frac{1}{n}\mathbf{I}_t\right)\,, \quad (16)$$

where $\mathbf{I}_t = \mathrm{Cov}[g(\boldsymbol{\theta}_t;\mathbf{x})]$ is the covariance of the gradient at $\boldsymbol{\theta}_t$. Given that the noisy (stochastic) gradient based on current subset $\nabla\tilde{U}(\boldsymbol{\theta}_t) = -\frac{N}{n}\sum_{i=1}^n g(\boldsymbol{\theta}_t;\mathbf{x}_{r_i}) - \nabla\log\pi(\boldsymbol{\theta}_t)$, and the clean (full)

---

**Algorithm 1** Covariance-Controlled Adaptive Langevin (CCAdL)

---

1: **Input:** $h$, $A$, $\{\kappa_t\}_{t=1}^{\hat{T}}$.
2: **Initialize** $\boldsymbol{\theta}_0$, $\mathbf{p}_0$, $\hat{\mathbf{I}}_0$, and $\xi_0 = A$.
3: **for** $t = 1, 2, \ldots, \hat{T}$ **do**
4:     $\boldsymbol{\theta}_t = \boldsymbol{\theta}_{t-1} + \mathbf{p}_{t-1} h$;
5:     Estimate $\hat{\mathbf{I}}_t$ using Eq. (18);
6:     $\mathbf{p}_t = \mathbf{p}_{t-1} - \nabla\tilde{U}(\boldsymbol{\theta}_t)h - \frac{h}{2}\frac{N^2}{n}\hat{\mathbf{I}}_t\mathbf{p}_{t-1}h - \xi_{t-1}\mathbf{p}_{t-1}h + \sqrt{2Ah}\mathcal{N}(0,1)$;
7:     $\xi_t = \xi_{t-1} + \left(\mathbf{p}_t^T\mathbf{p}_t/N_{\mathrm{d}} - 1\right)h$;
8: **end for**

---

gradient $\nabla U(\boldsymbol{\theta}_t) = -\sum_{i=1}^{N} g(\boldsymbol{\theta}_t; \mathbf{x}_i) - \nabla\log\pi(\boldsymbol{\theta}_t)$, we have $\mathbb{E}_{\mathbf{x}}[\nabla\tilde{U}(\boldsymbol{\theta}_t)] = \mathbb{E}_{\mathbf{x}}[\nabla U(\boldsymbol{\theta}_t)]$, and thus

$$\nabla\tilde{U}(\boldsymbol{\theta}_t) = \nabla U(\boldsymbol{\theta}_t) + \mathcal{N}\left(\mathbf{0}, \frac{N^2}{n}\mathbf{I}_t\right), \tag{17}$$

i.e. $\boldsymbol{\Sigma}(\boldsymbol{\theta}_t) = N^2\mathbf{I}_t/n$. Assuming $\boldsymbol{\theta}_t$ does not change dramatically over time, we use the moving average update to estimate $\mathbf{I}_t$,

$$\hat{\mathbf{I}}_t = (1 - \kappa_t)\hat{\mathbf{I}}_{t-1} + \kappa_t\mathbf{V}(\boldsymbol{\theta}_t), \tag{18}$$

where $\kappa_t = 1/t$, and

$$\mathbf{V}(\boldsymbol{\theta}_t) = \frac{1}{n-1}\sum_{i=1}^{n} \left(g(\boldsymbol{\theta}_t; \mathbf{x}_{r_i}) - \bar{g}(\boldsymbol{\theta}_t)\right)\left(g(\boldsymbol{\theta}_t; \mathbf{x}_{r_i}) - \bar{g}(\boldsymbol{\theta}_t)\right)^T \tag{19}$$

is the empirical covariance of gradient. $\bar{g}(\boldsymbol{\theta}_t)$ represents the mean gradient of the log likelihood computed from a subset. As proved in [2], this estimator has a convergence order of $O(1/N)$.

As already mentioned, estimating the full covariance matrix is computationally infeasible in high dimension. However, we have found that employing a diagonal approximation of the covariance matrix (i.e. only estimating the variance along each dimension of the noisy gradient), works quite well in practice, as demonstrated in Section 4.

The procedure of the CCAdL method is summarized in Algorithm 1, where we simply used $\mathbf{M} = \mathbf{I}$, $\beta = 1$, and $\mu = N_{\mathrm{d}}$ in order to be consistent with the original implementation of SGNHT [5].

Note that this is a simple, first order (in terms of the stepsize) algorithm. A recent article [15] has introduced higher order of accuracy schemes which can improve accuracy, but our interest here is in the direct comparison of the underlying machinery of SGHMC, SGNHT, and CCAdL, so we avoid further modifications and enhancements related to timestepping at this stage.

In the following section, we compare the newly-established CCAdL method with SGHMC and SGNHT on various machine learning tasks to demonstrate the benefits of CCAdL in Bayesian sampling with a noisy gradient.

## 4 Numerical Experiments

### 4.1 Bayesian Inference for Gaussian Distribution

We first compare the performance of the newly-established CCAdL method with SGHMC and SGNHT for a simple task using synthetic data, i.e. Bayesian inference of both the mean and variance of a one-dimensional normal distribution. We apply the same experimental setting as in [5]. We generated $N = 100$ samples from the standard normal distribution $\mathcal{N}(0, 1)$. We used the likelihood function of $\mathcal{N}(\mathbf{x}_i|\mu, \gamma^{-1})$ and assigned Normal-Gamma distribution as their prior distribution, i.e. $\mu, \gamma \sim \mathcal{N}(\mu|0, \gamma)\mathrm{Gam}(\gamma|1, 1)$. Then the corresponding posterior distribution is another Normal-Gamma distribution, i.e. $(\mu, \gamma)|\mathbf{X} \sim \mathcal{N}(\mu|\mu_N, (\kappa_N\gamma)^{-1})\mathrm{Gam}(\gamma|\alpha_N, \beta_N)$, with

$$\mu_N = \frac{N\bar{\mathbf{x}}}{N+1}, \qquad \kappa_N = 1 + N, \qquad \alpha_N = 1 + \frac{N}{2}, \qquad \beta_N = 1 + \sum_{i=1}^{N} \frac{(\mathbf{x}_i - \bar{\mathbf{x}})^2}{2} + \frac{N\bar{\mathbf{x}}^2}{2(1+N)},$$

where $\bar{\mathbf{x}} = \sum_{i=1}^{N} \mathbf{x}_i/N$. A random subset of size $n = 10$ was selected at each timestep to approximate the full gradient, resulting in the following stochastic gradients,

$$\nabla_\mu\tilde{U} = (N+1)\mu\gamma - \frac{\gamma N}{n}\sum_{i=1}^{n}\mathbf{x}_{r_i}, \qquad \nabla_\gamma\tilde{U} = 1 - \frac{N+1}{2\gamma} + \frac{\mu^2}{2} + \frac{N}{2n}\sum_{i=1}^{n}(\mathbf{x}_{r_i} - \mu)^2.$$

It can be seen that the variance of the stochastic gradient noise is no longer constant and actually depends on the size of the subset, $n$, and the values of $\mu$ and $\gamma$ in each iteration. This directly violates the constant noise variance assumption of SGNHT [5], while CCAdL adjusts to the varying noise variance.

The marginal distributions of $\mu$ and $\gamma$ obtained from various methods with different combinations of $h$ and $A$ were compared and plotted in Figure 1, with Table 1 consisting of the corresponding root mean square error (RMSE) of the distribution and autocorrelation time from $10^6$ samples. In most of the cases, both SGNHT and CCAdL easily outperform the SGHMC method possibly due to the presence of the Nosé-Hoover device, with SGHMC only showing superiority with small values of $h$ and large value of $A$, neither of which is desirable in practice as discussed in Section 3. Between SGNHT and the newly-proposed CCAdL method, the latter achieves better performance in each of the cases investigated, highlighting the importance of the covariance control with parameter-dependent noise.

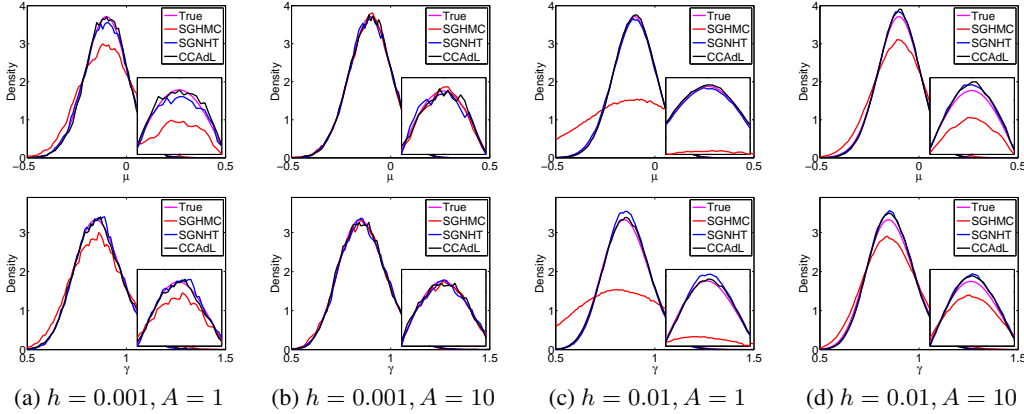

(a) $h = 0.001, A = 1$    (b) $h = 0.001, A = 10$    (c) $h = 0.01, A = 1$    (d) $h = 0.01, A = 10$

Figure 1: Comparisons of marginal distribution (density) of $\mu$ (top row) and $\gamma$ (bottom row) with various values of $h$ and $A$ indicated in each column. The peak region is highlighted in the inset.

Table 1: Comparisons of (RMSE, Autocorrelation time) of $(\mu, \gamma)$ of various methods for Bayesian inference of Gaussian mean and variance.

| Methods | $h = 0.001, A = 1$ | $h = 0.001, A = 10$ | $h = 0.01, A = 1$ | $h = 0.01, A = 10$ |
|---|---|---|---|---|
| SGHMC | $(0.0148, 236.12)$ | $(\mathbf{0.0029}, \mathbf{333.04})$ | $(0.0531, 29.78)$ | $(0.0132, 39.33)$ |
| SGNHT | $(0.0037, 238.32)$ | $(0.0035, 406.71)$ | $(0.0044, 26.71)$ | $(0.0043, 55.00)$ |
| CCAdL | $(\mathbf{0.0034}, \mathbf{238.06})$ | $(0.0031, 402.45)$ | $(\mathbf{0.0021}, \mathbf{26.71})$ | $(\mathbf{0.0035}, \mathbf{54.43})$ |

## 4.2 Large-scale Bayesian Logistic Regression

We then consider a Bayesian logistic regression model trained on the benchmark MNIST dataset for binary classification of digits 7 and 9 using $12,214$ training data points, with a test set of size 2037. A 100-dimensional random projection of the original features was used. We used the likelihood function of $\pi\left(\{\mathbf{x}_i, y_i\}_{i=1}^N | \mathbf{w}\right) \propto \prod_{i=1}^N 1 / \left(1 + \exp(-y_i \mathbf{w}^T \mathbf{x}_i)\right)$, and the prior distribution of $\pi(\mathbf{w}) \propto \exp(-\mathbf{w}^T \mathbf{w}/2)$, respectively. A subset of size $n = 500$ was used at each timestep. Since the dimensionality of this problem is not that high, a full covariance estimation was used for CCAdL.

We investigate the convergence speed of each method through measuring test log likelihood using posterior mean against the number of passes over the entire dataset, see Figure 2 (top row). CCAdL displays significant improvements over SGHMC and SGNHT with different values of $h$ and $A$: (1) CCAdL converges much faster than the other two, which also indicates its faster mixing speed and shorter burn-in period; (2) CCAdL shows robustness in different values of the "effective friction" $A$, with SGHMC and SGNHT relying on a relative large value of $A$ (especially the SGHMC method) which is intended to dominate the gradient noise.

To compare the sample quality obtained from each method, Figure 2 (bottom row) plots the two-dimensional marginal posterior distribution in randomly-selected dimensions of 2 and 5 based on $10^6$ samples from each method after the burn-in period (i.e. we start to collect samples when the

test log likelihood stabilizes). The true (reference) distribution was obtained by a sufficiently long run of standard HMC. We implemented 10 runs of standard HMC and found there was no variation between these runs, which guarantees its qualification as the true (reference) distribution. Again, CCAdL shows much better performance than SGHMC and SGNHT. Note that the SGHMC does not even fit in the region of the plot, and in fact it shows significant deviation even in the estimation of the mean.

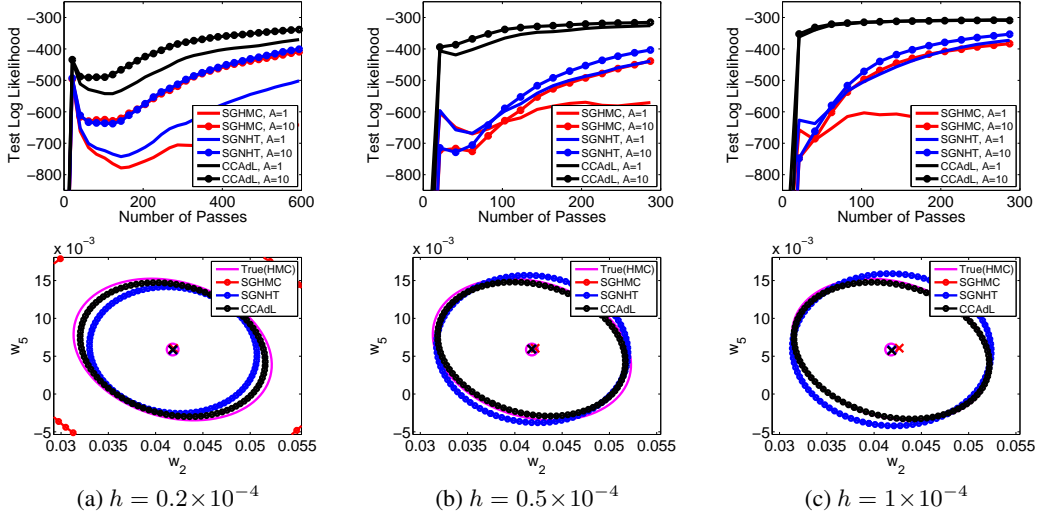

(a) $h = 0.2 \times 10^{-4}$  (b) $h = 0.5 \times 10^{-4}$  (c) $h = 1 \times 10^{-4}$

Figure 2: Comparisons of Bayesian Logistic Regression of various methods on the MNIST dataset of digits 7 and 9 with various values of $h$ and $A$: (top row) test log likelihood using posterior mean against number of passes over the entire dataset; (bottom row) two-dimensional marginal posterior distribution in (randomly selected) dimensions 2 and 5 with $A = 10$ fixed, based on $10^6$ samples from each method after the burn-in period (i.e. we start to collect samples when the test log likelihood stabilizes). Magenta circle is the true (reference) posterior mean obtained from standard HMC, and crosses represent the sample means computed from various methods. Ellipses represent iso-probability contours covering 95% probability mass. Note that the contour of SGHMC is well beyond the scale of figure and thus we do not include it here.

## 4.3   Discriminative Restricted Boltzmann Machine (DRBM)

DRBM [11] is a self-contained non-linear classifier, and the gradient of its discriminative objective can be explicitly computed. Due to the limited space, we refer the readers to [11] for more details. We trained a DRBM on different large-scale multi-class datasets from LIBSVM[1] dataset collection, including *connect-4*, *letter*, and *SensIT Vehicle acoustic*. The detailed information of these datasets are presented in Table 2.

We selected the number of hidden units using cross-validation to achieve their best results. Since the dimension of parameters, $N_d$, is relatively high, we only used diagonal covariance matrix estimation for CCAdL to significantly reduce the computational cost, i.e. only estimating the variance along each dimension. The size of the subset was chosen as 500–1000 to obtain a reasonable variance estimation. For each dataset, we chose the first 20% of the total number of passes over the entire dataset as the burn-in period, and collected the remaining samples for prediction.

Table 2: Datasets used in DRBM with corresponding parameter configurations.

| Datasets | training/test set | classes | features | hidden units | total number of parameters $N_d$ |
|---|---|---|---|---|---|
| *connect-4* | 54,046/13,511 | 3 | 126 | 20 | 2603 |
| *letter* | 10,500/5,000 | 26 | 16 | 100 | 4326 |
| *acoustic* | 78,823/19,705 | 3 | 50 | 20 | 1083 |

The error rate computed by various methods on the test set using posterior mean against number of passes over entire dataset was plotted in Figure 3. It can be observed that SGHMC and SGNHT only work well with a large value of the effective friction $A$, which corresponds to a strong random walk effect and thus slows down the convergence. On the contrary, CCAdL works reliably (much better than the other two) in a wide range of $A$, and more importantly in the large stepsize regime, which

speeds up the convergence rate in relation to the computational work performed. It can be easily seen that the performance of SGHMC heavily relies on using a small value of $h$ and large value of $A$, which significantly limits its usefulness in practice.

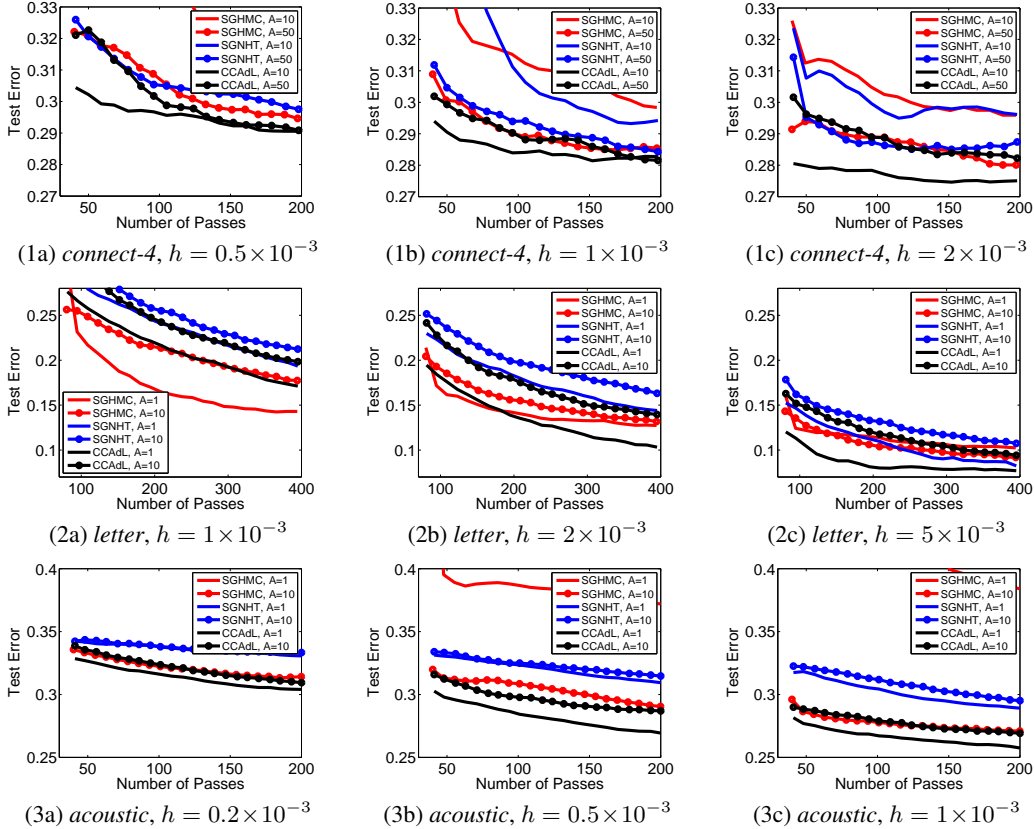

Figure 3: Comparisons of DRBM on datasets *connect-4* (top row), *letter* (middle row), and *acoustic* (bottom row) with various values of $h$ and $A$ indicated: test error rate of various methods using posterior mean against number of passes over the entire dataset.

## 5 Conclusions and Future Work

In this article, we have proposed a novel Covariance-Controlled Adaptive Langevin (CCAdL) formulation that can effectively dissipate parameter-dependent noise while maintaining a desirable invariant distribution. CCAdL combines ideas of SGHMC and SGNHT from the literature, but achieves significant improvements over each of these methods in practice. The additional error introduced by covariance estimation is expected to be small in a relative sense, i.e. substantially smaller than the error arising from the noisy gradient. Our findings have been verified in large-scale machine learning applications. In particular, we have consistently observed that SGHMC relies on a small stepsize $h$ and large friction $A$, which significantly reduces its usefulness in practice as discussed. The techniques presented in this article could be of use in the more general setting of large-scale Bayesian sampling and optimization, which we leave for future work.

A naive nonsymmetric splitting method has been applied for CCAdL for fair comparison in this article. However, we point out that optimal design of splitting methods in ergodic SDE systems has been explored recently in the mathematics community [1, 13, 14]. Moreover, it has been shown in [15] that a certain type of symmetric splitting method for the Ad-Langevin/SGNHT method with a clean (full) gradient inherits the superconvergence property (i.e. fourth order convergence to the invariant distribution for configurational quantities) recently demonstrated in the setting of Langevin dynamics [12, 14]. We leave further exploration of this direction in the context of noisy gradients for future work.

## Footnotes

[1]http://www.csie.ntu.edu.tw/~cjlin/libsvmtools/datasets/multiclass.html

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
