[Reviews · NeurIPS 2015]

Submitted by Assigned_Reviewer_1

A covariance controlled Nose Theromstat is presented to sample from a target measure when computing the likelihhod is infeasible due to far too many terms appearing in the product form likelihood.

This paper builds on much of the solid work in molecular dynamics by the likes of Leimkuhler and is an interesting addition to this emerging area of literature.

The evaluations experimentally are a bit simplistic thought the final DBM was interesting.

Quality This is a solid piece of work that takes results from molecular dynamics and uses them to address a contemporary ML problem.

Clarity The paper is clearly written and there was no level of unnecessary complication or obfuscation - a very clear and accessible read.

Originality A similar construction - with fixed covariance - appeared at last years NIPS - and this builds and improves on that work by Skeel. I would not take this as an impediment of the paper as it is a valuable addition.

Significnace This area is going to be significant - for models with factorable likelihoods - a limited class of statistical models
Summary: A covariance controlled Nose Theromstat is presented to sample from a target measure when computing the likelihhod is infeasible due to far too many terms appearing in the product form likelihood.

This paper builds on much of the solid work in molecular dynamics by the likes of Leimkuhler and is an interesting addition to this emerging area of literature.

The evaluations experimentally are a bit simplistic thought the final DBM was interesting.

Submitted by Assigned_Reviewer_2

This paper proposed the CCAdL method, which improves over the SGNHT method when the variance of stochastic gradients is not constant over the whole parameter space.

The cost is that the CCAdL method has to explicitly estimate the stochastic gradient variance. The estimation of the SG variance can be conducted on the minibatch, which is reasonably efficient. But this introduces an additional noise which comes from this estimator itself (acknowledged by the author in line 193). Although the thermostats can stabilize the system by neutralize constant noises, the noise from this estimator is clearly also parameter dependent.

Therefore, at the end of day, there is still some error in the system which cannot be removed just like the SGNHT. But based on the good experimental result, it may be possible that the impact of this error is not as severe as the original error from the SG. It would be very interesting, if the authors could look into the problem and characterize this error compared with the error from the SG.

The paper is well written and the experimental results look good (although only small scale experiments are conducted). But without a more careful analysis about the error of the new introduced stochastic noise, the paper may be incremental in terms of the overall novelty.
Summary: The paper incorporate an estimator of the variance of the stochastic gradients into the SGNHT. The algorithm is incremental over SGNHT and it has a flaw which has not been fully addressed.

Submitted by Assigned_Reviewer_3

The paper introduces covariance-controlled adaptive Langevin thermostat (CCAdL), a Bayesian sampling method based on stochastic gradients (SG) that aims to account for correlated errors introduced by the SG approximation of the true gradient. The authors demonstrate that CCAdL is more accurate and robust than other SG based methods on various test problems.

In general, the paper is well written but sometimes a bit hard to follow for someone who is not familiar with these type of sampling algorithms. The paper starts by reviewing various SG methods for efficient Bayesian posterior sampling (SGDL, mSGDL, SGHMC, SGHNT). It would be quite helpful if the authors could provide, for example, a table or figure that gives on overview over the different SG variants and highlights their commonalities and differences.

CCAdL combines ideas that have been proposed previously (mainly SGHMC and SGNHT):

- approximation of the true gradient of the log posterior with a stochastic version obtained by sub-sampling the data; this introduces "noise" in the gradient which has to be dealt with

- an estimate of the covariance matrix of the gradient noise; the estimate is a running average over the Fisher scores; in high-dimensional problems the authors replace it by the diagonal matrix

- the use of a thermostat in order to account for the inefficiency of Metropolis Monte Carlo acceptance/rejection used, e.g., in standard HMC

Given that I'm not an expert in the field, I'm not sure what the real novelty of CCAdL is. It apparently combines ideas from SGHMC and SGHNT with a previously proposed estimator for the noise covariance matrix.

The authors demonstrate CCAdL on a logistic regression problem and show that it converges significantly faster to higher log likelihood values. CCAdL also works for friction values smaller than those needed by SGHMC and SGHNT to result in stable sampling. By looking at the marginal distribution of pairs of parameters, the authors show that CCAdL produces posterior distributions that are close to the "true" distribution obtained by HMC on the full likelihood. As a second large scale example the authors train and test discriminative RBMs on three datasets. Again, they find that CCAdL performs superior to SGHMC and SGHNT for most stepsizes and friction constants.

These results are quite promising and deserve publication. It would be nice if the authors could improve the paper in terms of readability for the non-expert and correct some details. For example, some symbols are not explained when they are introduced, e.g. $\mu$ and $dW_A$.
Summary: The paper introduces covariance-controlled adaptive Langevin thermostat (CCAdL) a Bayesian sampling method that is based on stochastic gradients (SG) and aims to account for correlated errors introduced by the SG approximation of the true gradient. The authors demonstrate that CCAdL is more accurate and robust than other SG based methods.

Submitted by Assigned_Reviewer_4

Paper Title: Covariance-Controlled Adaptive Langevin Thermostat for Large-Scale Bayesian Sampling

Paper Summary: This paper presents a new method (the "covariance-controlled adaptive Langevin thermostat") for MCMC posterior sampling for Bayesian inference. Along the lines of previous work in scalable MCMC, this is a stochastic gradient sampling method. The presented method aims to decrease parameter-dependent noise (in order to speed-up convergence to the given invariant distribution of the Markov chain, and generate beneficial samples more efficiently), while maintaining the desired invariant distribution of the Markov chain. Similar to existing stochastic gradient MCMC methods, this method aims to find use in large-scale machine learning settings (i.e. Bayesian inference with large numbers of observations). Experiments on three models (a normal-gamma model, Bayesian logistic regression, and a discriminative restricted Boltzmann machine) aim to show that the presented method performs better than stochastic gradient Hamiltonian monte carlo (SGHMC) and stochastic gradient Nose-Hoover thermostat (SGNHT), two similar existing methods.

Comments:

- I feel that this paper proposes a valid contribution to the area of stochastic gradient MCMC methods, and does a good job putting this method in context with similar previous methods (SGHMC and SGNHT). However, one detriment of this paper is that it is somewhat incremental, both in terms of ideas and the results shown.

- In the experiments, comparisons are only against two methods: SGHMC and SGNHT. It might be nice to also see results for some of the other recently developed mini-batch MCMC methods, (such as the original stochastic gradient Langevin dynamics, or stochastic gradient Fisher scoring), or for some of the methods that do not rely on stochastic gradients, such as: (Bardenet, Remi, Arnaud Doucet, and Chris Holmes. "On Markov chain Monte Carlo methods for tall data." arXiv preprint arXiv:1505.02827 (2015)), or (Maclaurin, Dougal, and Ryan P. Adams. "Firefly Monte Carlo: Exact MCMC with subsets of data." arXiv preprint arXiv:1403.5693 (2014)).

- In Figure 1, the inset "peaks" add very little to the figure -- they seem to be an only very-slight zoom into what is shown in the non-inset part of the figure.

- I feel there are a few places in this paper where the quality of writing could be improved. In the abstract, there are few sentences that feel somewhat ambiguous to me (such as "one area of current research asks how to utilise the computational benefits of stochastic gradient methods in this setting."). In the intro (second paragraph), the order of presentation of stochastic gradient methods seems odd (first, the collection of all existing methods are described, and then afterwards, the first developed method is described). In section 2, there is a bit of confusion when a few terms are introduced without enough description (such as "temperature", "Boltzmann constant"); it would be better to give a brief description or intuition when introducing these terms to a machine learning audience.
Summary: I feel that developing better methods for scalable Bayesian inference is important, and that this paper does a good job of combining benefits from two similar methods, SGHMC and SGNHT, and showing better performance in practice. However, I feel that the contributions made by this paper are somewhat incremental, and that more care could be taken to show results with other recently developed comparison methods.

Author Feedback
Author rebuttal: We wish to thank the reviewers for their valuable suggestions and comments.

Reviewer 2:

The reviewer notes that the covariance estimation introduces a noise that is not corrected in the proposed formulation.

Response:

The additional error introduced by covariance estimation is expected to be substantially smaller than the original error. The technique proposed thus represents an improvement on the existing methodology. Residual error is automatically and adaptively handled using the Nose-Hoover device. Numerical experiments included in the paper demonstrate that the newly-proposed CCAdL method can significantly reduce the error and drive the system more rapidly into thermodynamic equilibrium. The combination of covariance estimation and adaptive correction reliably outperforms the SGHMC and SGNHT methods from the literature.

Reviewer 3:

The reviewer requests some improvement in the presentation for nonspecialists and asks that details be provided on the definitions of all parameters.

Response:

We will expand the introduction somewhat to make the paper more accessible. We have added definitions of $\mu$ and $dW_A $ following the formulation of CCAdL and will ensure that no other parameters are introduced without description.

Reviewer 4:

While recommending publication, the reviewer suggests that the benefit of the CCAdL method is ``somewhat incremental''. The reviewer suggests additional numerical experiments with certain specified MCMC schemes.

Response:

Our numerical experiments indicate that the CCAdL method is not by any means an incremental improvement, but potentially, at least for applications such as those we have considered, {\em orders of magnitude more efficient} (see Figures 2 and 3) than the SGHMC and SGNHT methods, which are the state-of-the-art schemes for addressing stochastically perturbed gradient forces in machine learning applications. (The mentioned MCMC methods, while certainly interesting, are not the right comparisons for our method.)

We will modify the abstract and introduction to highlight the dramatic improvement in efficiency obtained using the CCAdL scheme.

Reviewers 5-7 are highly favorable and provide no negative feedback to address.